Bioinformatical analysis identifies PDLIM3 as a potential biomarker associated with immune infiltration in patients with endometriosis

Gan Lei
Sun Jiani
Sun Jing sunjing61867@163.com
Shanghai First Maternity and Infant Hospital, School of Medicine, Tongji University , Shanghai , China
Polimanti Renato
Electronic publication date: 2022 Mar 30
Publication date: 2022
Volume: 10
Electronic Location ID: e13218
Received 2022 Jan 21; Accepted 2022 Mar 14
Copyright: © 2022 Gan et al.
Copyright year: 2022
Copyright holder: Gan et al.
License: This is an open access article distributed under the terms of the Creative Commons Attribution License, which permits unrestricted use, distribution, reproduction and adaptation in any medium and for any purpose provided that it is properly attributed. For attribution, the original author(s), title, publication source (PeerJ) and either DOI or URL of the article must be cited.
License URL: https://creativecommons.org/licenses/by/4.0/

Keywords: PDLIM3, Endometriosis, Diagnostic gene biomarker, Immune infiltration, bioinformatic, Machine learning

Funding: Shanghai Science and Technology Committee 19411960300 Shanghai Hospital Development Center SHDC12019113 This work was supported by the Shanghai Science and Technology Committee under grant number 19411960300 to Jing Sun, and the Shanghai Hospital Development Center under grant number SHDC12019113 to Jing Sun. The funders had no role in study design, data collection and analysis, decision to publish, or preparation of the manuscript.

==============================
Background

Endometriosis is a chronic systemic disease, whose classic symptoms are pelvic pain and infertility. This disease seriously reduces the life quality of patients. The pathogenesis, recognition and treatment of endometriosis is still unclear, and cannot be over emphasized. The aim of our study was to investigate the potential biomarker of endometriosis for the mechanism and treatment.

Methods

Using GSE11691, GSE23339 and GSE5108 datasets, differentially expressed genes (DEGs) were identified between endometriosis and normal samples. The functions of DEGs were reflected by the analysis of gene ontology (GO), pathway enrichment and gene set enrichment analysis (GSEA). The LASSO regression model was performed to identify candidate biomarkers. The receiver operating characteristic curve (ROC) was used to evaluate discriminatory ability of candidate biomarkers. The predictive value of the markers in endometriosis were further validated in the GSE120103 dataset. Then, the expression level of biomarkers was detected by qRT-PCR and Western blot. Finally, the relationship between candidate biomarker expression and immune infiltration was estimated using CIBERSORT.

Results

A total of 42 genes were identified, which were mainly involved in cytokine–cytokine receptor interaction, systemic lupus erythematosus and chemokine signaling pathway. We confirmed PDLIM3 was a specific biomarker in endometriosis (AUC = 0.955) and validated in the GSE120103 dataset (AUC = 0.836). The mRNA and protein expression level of PDLIM3 in endometriosis tissue was significantly higher than normal. Immune cell infiltration analysis revealed that PDLIM3 was correlated with M2 macrophages, neutrophils, CD4+ memory resting T cells, gamma delta T cells, M1 Macrophages, resting mast cells, follicular helper T cells, activated NK cells, CD8+ T cells, regulatory T cells (Tregs), naive B cells, plasma cells and resting NK cells.

Introduction

Recently, endometriosis is defined as a chronic systemic disease, characterized by the presence and infiltration of ectopic endometrial tissue (Taylor, Kotlyar & Flores, 2021). It affects nearly 10% of women who are of reproductive age. The most common symptoms are pelvic pain and infertility. Nearly 70% of teens with pelvic pain are later diagnosed with endometriosis (Yeung et al., 2011). Endometriosis seriously reduces the life quality of patients, increases incidence of depression and causes a heavy burden on the direct and indirect healthcare (Agarwal et al., 2019). The gold standard for endometriosis diagnosis is invasive laparoscopy examination. It is difficult to recognize because of patients’ nonspecific descriptions of symptoms, especially at early stages. Despite its prevalence, the average time from symptom onset to diagnosis is 5 to 10 years (Hudelist et al., 2012; Greene et al., 2009). The new opinion about diagnosis of endometriosis is emphasize disease symptoms and their origins rather than lesion presence or absence (Yeung et al., 2011). Specific circulating biomarkers, such as miR-125b-5p, miR-150-5p, miR-61, have been studied as promising early clinical diagnostic tools, although functional researches are needed to confirm their effect on this disease. Currently, medical therapy and surgical resection are the gold standard treatment for endometriosis (Vercellini et al., 2014) while both are associated with high recurrence rates and limited by high occurrence of side effects. The mechanisms underlying the pathogenesis and development of endometriosis will open the door to a new approach to treatment.

Various hypotheses have been proposed to explain the underlying molecular mechanisms of endometriosis, consisting of the endometrial implantation theory of Sampson (Kruitwagen et al., 1991), eutopic endometrium determinism, stem cell factors (Burney & Giudice, 2012), epigenetic factors, immunodeficiency as well as immune and inflammatory factors (Ahn et al., 2015).

Recent clinical and molecular studies have suggested that innate properties or acquired properties of the endometrium and impaired immune homeostasis are systems of significance in explaining the mechanism of endometriosis (Burney & Giudice, 2012). Immune imbalance is associated with increased implantation, proliferation, and angiogenesis of ectopic endometrium. Previous studies have shown that immune dysfunction happens in peritoneal fluid of endometriosis, which may contribute to the progression of endometriosis. Functional changes in immune components have been indicated, consisting of natural killer (NK) cells, monocytes/macrophages, B cells, T lymphocytes, and cytokines (Zou et al., 2021). Although numerous studies have described occurrence of immune abnormalities in endometriosis, the role of the immune system is not clearly understood. Hence, it is essential to explore mechanism of immune dysfunction in endometriosis, which can contribute to elucidate the role of immune system in pathogenesis of endometriosis and generate applicable insights to develop preventive and control strategies, original non-invasive diagnostic methods and targeted therapeutics.

In the present study, we downloaded three microarray datasets (GSE11691, GSE23339, GSE5108) from the Gene Expression Omnibus (GEO) database to detect the potential biomarkers for further analysis. We identified candidate genes, explore immune mechanism and key signaling pathways regulating occurrence, development and progression of endometriosis through bioinformatics method. In conclusion, these findings provide novel insights into understanding the underlying immune mechanisms and seeking for potential molecular treatment.

Materials and Methods

Gene expression profile

We searched the associated gene expression profiles of endometriosis in NCBI-GEO (https://www.ncbi.nlm.nih.gov/geo/) (Barrett et al., 2005), using the keywords “endometriosis” and “Homo sapiens”. Three gene expression datasets containing microarray data from normal endometrial tissues and ectopic endometrial tissues, GSE11691 (Hull et al., 2008), GSE23339 (Hawkins et al., 2011) and GSE5108 (Eyster et al., 2007), were selected for further analysis. The GSE11691 microarray data consisted of nine normal human endometrium and nine ectopic human endometrium (two from the peritoneal wall, seven from the broad ligament) and was generated using the GPL96 platform Affymetrix Human Genome U133A Array. The GSE23339 microarray data consisted of nine normal human endometrium and 10 ectopic human endometrium (all from the endometriotic cysts of the ovaries) and was generated using the GPL6102 Illumina human-6 v2.0 expression beadchip. The GSE5108 microarray data consisted of 11 normal human endometrium and 11 ectopic human endometrium (five from the peritoneal wall, six from the endometriotic cysts of the ovaries) and was generated using the GPL2895 GE Healthcare/Amersham Biosciences CodeLink Human Whole Genome Bioarray. In addition, the GSE120103 (Bhat et al., 2019) dataset, containing 18 endometriosis (all from the endometriotic cysts of the ovaries) and 18 control samples, was used as the validation cohort using the GPL6480 platform Agilent-014850 Whole Human Genome Microarray 4 × 44K G4112F. GSE12768 (Ritchie et al., 2015) dataset, containing two endometriosis (both from the endometriotic cysts of the ovaries) and two control samples, was used as another validation cohort using the GPL7304 Institut Cochin HG18 60mer expression array 47K.

Identification of differentially expressed genes (DEGs)

The raw data of three datasets, GSE11691, GSE23339 and GSE5108, was merged and removed batch effects by “sva” package (http://www.bioconductor.org/) (Ritchie et al., 2015). Then, the “limma” package (http://www.bioconductor.org/) was used to attain DEGs between endometriosis samples and normal samples. The raw data of GSE12768 was analyzed separately to acquired the DEGs list, and compared with the list of three datasets in order to verify the reliability of our analysis. The significant DEGs were identified according to adjusted P value < 0.05 and ∣log fold change (FC)∣ > 2.

Functional and pathway enrichment analysis of DEGs

To reflect gene functions, gene ontology (GO) (http://geneontology.org/) (Thomas, 2017) biological processes were shown in three aspects: biological processes (BP), cellular component (CC) and molecular function (MF). And the analysis process was achieved by “clusterProfiler” package (Wu et al., 2021; Yu et al., 2012) in the present study, using the cut-off criterion P value < 0.05. ToppGene (https://toppgene.cchmc.org/enrichment.jsp) (Chen et al., 2009) is a free tool for uncovering the biological significance behind a cluster of genes. The REACTOME (https://reactome.org/) (Jassal et al., 2020) pathway enrichment analysis, was exactly performed using this tool. A P value < 0.05 was considered as the threshold points.

Moreover, to get more information of mechanism behind DEGs, gene set enrichment analysis (GSEA) was employed to explore the potential feature between the endometriosis and control groups. The “c2.cp.kegg.v7.4.symbols.gmt” from the Molecular Signatures Database (MSigDB) was downloaded and used as the reference gene set. An adjusted P < 0.05 and false discovery rate (FDR) < 0.05 were considered as significant.

Protein–protein interaction (PPI) networks construction

STRING (https://string-db.org/) (Szklarczyk et al., 2015) is a tool designed to achieve a comprehensive and objective global network and visualize PPI information. Proteins associated with DEGs were chosen based on information in STRING (confidence score > 0.4), and then PPI networks were constructed using Cytoscape software (http://cytoscape.org/) (Shannon et al., 2003).

Screening of specific gene biomarkers

The least absolute shrinkage and selection operator (LASSO), a regression analysis algorithm, was applied to evaluate the value of DEGs in this study. The “glmnet” package in R did the above analysis and returned candidate potential genes. And the expression levels of candidate biomarkers were further tested in the GSE120103 dataset.

Receiver operating characteristic (ROC) curve analysis

To test the predictive accuracy of the identified biomarkers, we generated ROC curves using “pROC” package (Robin et al., 2011) based on the candidate genes and compare the predictive value of candidate genes by the area under the ROC curve (AUC). And then, the predictive value of these genes was also validated in the GSE120103 dataset.

Relationship between identified genes and infiltrating immune cells in endometriosis

We used the CIBERSORT (https://cibersortx.stanford.edu/) (Newman et al., 2015; Robin et al., 2011) to calculate immune cell infiltrations of endometriosis and control tissues. Then, we applied “corrplot” and “vioplot” R package to visualize the results and explore correlation of infiltrating immune cells. In addition, the relation of the candidate gene markers with the levels of infiltrating immune cells was calculated and visualized by “ggpubr” package.

Quantitative real-time PCR (qRT-PCR)

Total RNA was isolated from ovarian endometriosis and eutopic endometrium (NC). These operations were approved by Ethics committee of Shanghai First Maternity and Infant Hospital (KS21198). We used Trizol (RNAiso Plus; Takara, Tokyo, Japan) to isolate RNA from these two kinds of tissues according to the manufacturer’s instructions. Then, one microgram of RNA was reversely transcribed into cDNA using a PrimeScript™ RT reagent kit (Takara, Tokyo, Japan). Amplification was performed using gene-specific primers (Sangon, Shanghai, China) and TB Green® Premix EX TaqTM II (Takara, Tokyo, Japan) on a qRT-PCR device (QuantStudio5, Thermo Fisher Scientific, Waltham, MA, USA). β-actin was used as an internal control. The relative expression of the genes was calculated using the 2–ΔΔCT method. The primers used were as follows: human β-actin-Forward: 5′-GTGCTATGTTGCTCTAGACTTCG-3′; human β-actin-Reverse: 5′-ATGCCACAGGATTCCATACC-3′; human PDLIM3-Forward: 5′-GAGTCGGACGTGTACCGGA-3′; human PDLIM3-Reverse: 5′-TGCCACTCCCACATTTGTCAC-3′.

Protein extraction and western blot

Normal endometrium tissues and ovarian endometriosis tissues were lysed in RIPA lysis buffer (PC101; EpiZyme, Shanghai, China) with protease and phosphatase inhibitor cocktail. The protein concentration was detected by a BCA protein assay. The same amounts of total protein (32 µg) were loaded and separated onto 12.5% SDS-PAGE and was then transferred to PVDF membranes. After blocking with protein-free rapid blocking buffer (PS108P; EpiZyme, Shanghai, China) for 0.5 h at room temperature, the membranes were incubated with primary antibodies at 4 °C overnight. The primary antibodies rabbit anti-PDLIM3 antibody (1:2,000; Servicebio) was used and rabbit anti-β-Actin (1:5,000; ABclonal) served as the internal control.

The membranes were probed with secondary antibodies (1:3,000, Servicebio) for 1.5 h at room temperature and three washes in TBST were processed to remove excess secondary antibody. We then used an ECL western blotting kit (NCM Biotech, Suzhou, China) to visualize and image the targeted protein bands.

Statistical analysis

Most statistical calculations were conducted using R (version 4.1.1). The results of qRT-PCR and western blot were shown as the mean ± SD and GraphPad Prism 8.0 (GraphPad Software, San Diego, CA, USA) were used for statistical analysis. All the comparisons between two groups were analyzed using Student’s t-test for normally distributed variables. If the variables distribution was abnormal, the Mann–Whitney U-test would be used. The relationship between the expression of biomarkers and infiltrating immune cells was analyzed using Spearman’s rank correlation analysis. Two-sided with P < 0.05 (95% confidence interval) and false discovery rate (FDR) < 0.05 were considered as statistically significant.

Results

Identification of DEGs in endometriosis

Data from a total of 30 endometriosis samples (14 peritoneal endometriosis samples, 16 ovarian endometriosis samples) and 29 control endometrium samples from three GEO datasets (GSE11691, GSE23339 and GSE5108) were analyzed. The DEGs were calculated using the “limma” package after removing the batch effects. We obtained a total of 42 DEGs, including upregulated 23 genes and downregulated 19 genes (Table 1). All DEGs were shown in the heatmap and volcano map (Fig. 1). In addition, the DEGs list from GSE12768 dataset was displayed in Table S1.

Figure 1 Differentially expressed genes (DEGs) between endometriosis tissues and normal endometrium.

(A) Heatmap of DEGs. (B) Volcano Plot of DEGs.

Table 1 The statistical metrics for key differentially expressed genes (DEGs).

Gene symbol	logFC	P Value	adj. P Val	t value	Regulation	
C7	3.783999053	2.71E–13	1.35E–10	9.285264	Up	
PTGIS	3.604098022	3.16E–20	3.15E–16	13.64407	Up	
FABP4	3.468506892	3.05E–16	4.17E–13	11.06606	Up	
PLA2G2A	3.420962661	2.02E–10	2.06E–08	7.609885	Up	
MYH11	2.919064902	6.89E–10	5.46E–08	7.301042	Up	
WISP2	2.750733339	1.73E–10	1.81E–08	7.648918	Up	
CFH	2.682022262	2.63E–16	4.17E–13	11.10573	Up	
PDLIM3	2.630060076	1.29E–14	8.06E–12	10.07301	Up	
C3	2.621335795	2.33E–12	6.85E–10	8.735987	Up	
CLDN11	2.61447217	6.34E–13	2.53E–10	9.067731	Up	
PLN	2.549876617	2.77E–11	4.53E–09	8.110245	Up	
FZD7	2.50539562	2.88E–16	4.17E–13	11.08116	Up	
DPYSL3	2.334733121	2.79E–18	1.39E–14	12.35555	Up	
FMO1	2.287811528	1.46E–15	1.62E–12	10.64648	Up	
FAM129A	2.240230766	1.77E–14	1.04E–11	9.990994	Up	
S100A8	2.237119405	9.13E–10	6.96E–08	7.230211	Up	
LY96	2.195123038	3.34E–16	4.17E–13	11.04102	Up	
CPE	2.160535979	1.47E–12	4.75E–10	8.852691	Up	
COL8A1	2.148521844	7.63E–12	1.73E–09	8.435667	Up	
CXCL2	2.082497549	3.79E–06	5.78E–05	5.0803	Up	
CD163	2.06809014	2.63E–11	4.38E–09	8.12315	Up	
AOX1	2.026180382	1.98E–09	1.32E–07	7.034879	Up	
CCL2	2.006001247	2.29E–10	2.20E–08	7.578566	Up	
RORB	–2.07036228	2.91E–08	1.12E–06	–6.3538	Down	
TMPRSS4	–2.12081688	4.98E–09	2.79E–07	–6.80216	Down	
EHF	–2.13597521	7.95E–10	6.25E–08	–7.26498	Down	
SCNN1A	–2.14340132	7.39E–11	9.84E–09	–7.86305	Down	
MSX1	–2.17101832	9.74E–10	7.30E–08	–7.21392	Down	
KIAA1324	–2.18148615	1.16E–10	1.37E–08	–7.74893	Down	
ASRGL1	–2.24612274	3.73E–10	3.42E–08	–7.45546	Down	
PRSS8	–2.27282731	2.86E–11	4.61E–09	–8.10186	Down	
RAB25	–2.35255482	1.18E–11	2.35E–09	–8.32591	Down	
HSD17B2	–2.36471296	4.09E–07	9.60E–06	–5.67198	Down	
DEFB1	–2.38849343	3.57E–08	1.32E–06	–6.3015	Down	
GNLY	–2.44390133	1.59E–08	6.89E–07	–6.50697	Down	
MMP26	–2.60194621	5.19E–07	1.15E–05	–5.60914	Down	
AGR2	–2.62956901	1.19E–08	5.42E–07	–6.58042	Down	
CLDN3	–2.67438295	2.53E–10	2.41E–08	–7.55304	Down	
CD24	–2.68174109	6.68E–11	9.13E–09	–7.88849	Down	
GABRP	–2.84571026	1.07E–09	7.82E–08	–7.18967	Down	
SCGB1D2	–2.97903101	1.58E–05	0.000193	–4.68827	Down	
SCGB2A1	–3.6734424	5.33E–09	2.93E–07	–6.78496	Down	

GO and pathway enrichment of DEGs in endometriosis

The above list of DEGs was used to evaluate if they were enriched into specific GO or reactome pathways. GO analysis identified that the DEGs were significantly enriched in BP, including humoral immune response, complement activation, alternative pathway, antimicrobial humoral response, and so on (Table 2). In terms of MF, DEGs were mainly enriched in Toll-like receptor binding, G protein-coupled receptor binding, long-chain fatty acid binding, and chemokine receptor binding (Table 2). Moreover, CC demonstrated that filamentous actin, lipid droplet, sperm flagellum and actin filament were the most significantly enriched GO term (Table 2). The results were displayed in Figs. 2A and 2B. In addition, the GSEA results illustrated that the enriched pathways mainly involved cytokine–cytokine receptor interaction, systemic lupus erythematosus, chemokine signaling pathway, complement and coagulation cascades, and leishmania infection (Figs. 2C and 2D).

Figure 2 Functional enrichment analyses to identify potential biological processes.

(A, B) Gene ontology enrichment analysis of DEGs between endometriosis and control samples. (C, D) Enrichment analyses via gene set enrichment analysis.

Table 2 The enriched GO terms of differentially expressed genes.

ID	Category	GO Name	P Value	adj. P Val	Count	Gene ID	
GO:0006959	BP	Humoral immune response	4.84E–08	5.04E–05	9	CFH/C7/C3/PLA2G2A/CCL2/S100A8/GNLY/DEFB1/CXCL2	
GO:0006957	BP	Complement activation, alternative pathway	6.03E–06	0.003141	3	CFH/C7/C3	
GO:0019730	BP	Antimicrobial humoral response	9.26E–06	0.003216	5	PLA2G2A/S100A8/GNLY/DEFB1/CXCL2	
GO:0002237	BP	Response to molecule of bacterial origin	6.14E–05	0.015984	6	LY96/FMO1/CD24/CCL2/S100A8/CXCL2	
GO:0050727	BP	Regulation of inflammatory response	8.37E–05	0.01744	6	PTGIS/FABP4/C3/PLA2G2A/S100A8/MMP26	
GO:0050729	BP	Positive regulation of inflammatory response	0.000145	0.025149	4	FABP4/C3/PLA2G2A/S100A8	
GO:0097278	BP	Complement-dependent cytotoxicity	0.000176	0.026206	2	CFH/C3	
GO:0032496	BP	Response to lipopolysaccharide	0.00047	0.059867	5	LY96/FMO1/CCL2/S100A8/CXCL2	
GO:0061844	BP	Antimicrobial humoral immune response mediated by antimicrobial peptide	0.000517	0.059867	3	GNLY/DEFB1/CXCL2	
GO:0016338	BP	Calcium-independent cell-cell adhesion via plasma membrane cell-adhesion molecules	0.000974	0.092221	2	CLDN11/CLDN3	
GO:0031941	CC	Filamentous actin	0.001858	0.170931	2	DPYSL3/PDLIM3	
GO:0005811	CC	Lipid droplet	0.015451	0.170931	2	FABP4/CLDN11	
GO:0036126	CC	Sperm flagellum	0.020708	0.170931	2	SCNN1A/DEFB1	
GO:0005884	CC	Actin filament	0.02141	0.170931	2	DPYSL3/PDLIM3	
GO:0005923	CC	Bicellular tight junction	0.022846	0.170931	2	CLDN11/CLDN3	
GO:0005788	CC	Endoplasmic reticulum lumen	0.023003	0.170931	3	FMO1/C3/COL8A1	
GO:0097729	CC	9+2 motile cilium	0.023948	0.170931	2	SCNN1A/DEFB1	
GO:0070160	CC	Tight junction	0.025451	0.170931	2	CLDN11/CLDN3	
GO:1902495	CC	Transmembrane transporter complex	0.02645	0.170931	3	PLN/SCNN1A/GABRP	
GO:0005859	CC	Muscle myosin complex	0.029564	0.170931	1	MYH11	
GO:0035325	MF	Toll-like receptor binding	0.000258	0.014279	2	LY96/S100A8	
GO:0001664	MF	G protein-coupled receptor binding	0.000271	0.014279	5	FZD7/C3/CCL2/DEFB1/CXCL2	
GO:0036041	MF	Long-chain fatty acid binding	0.000305	0.014279	2	FABP4/S100A8	
GO:0042379	MF	Chemokine receptor binding	0.000362	0.014279	3	CCL2/DEFB1/CXCL2	
GO:0005504	MF	Fatty acid binding	0.002524	0.079747	2	FABP4/S100A8	
GO:0048020	MF	CCR chemokine receptor binding	0.003877	0.086681	2	CCL2/DEFB1	
GO:0005044	MF	Scavenger receptor activity	0.004045	0.086681	2	CD163/TMPRSS4	
GO:0008009	MF	Chemokine activity	0.004389	0.086681	2	CCL2/CXCL2	
GO:0033293	MF	Monocarboxylic acid binding	0.00902	0.158341	2	FABP4/S100A8	
GO:0038024	MF	Cargo receptor activity	0.010022	0.158341	2	CD163/TMPRSS4	

REACTOME pathway enrichment analysis was also used to obtain the signaling pathways for different genes. In this study, these DEGs were mainly involved in the regulation of complement cascade, suprofen pathway, indomethacin pathway and mefenamic acid pathway (Table 3).

Table 3 The enriched pathway terms of differentially expressed genes.

ID	Pathway name	P Value	Gene count	Gene ID	
1269250	Regulation of Complement cascade	4.35E–05	3	CFH, C3, C7	
SMP00101	Suprofen Pathway	1.34E–04	2	PLA2G2A, PTGIS	
SMP00104	Indomethacin Pathway	1.34E–04	2	PLA2G2A, PTGIS	
SMP00109	Mefanamic acid Pathway	1.34E–04	2	PLA2G2A, PTGIS	
SMP00113	Oxaprozin Pathway	1.34E–04	2	PLA2G2A, PTGIS	
SMP00114	Nabumetone Pathway	1.34E–04	2	PLA2G2A, PTGIS	
SMP00120	Naproxen Pathway	1.34E–04	2	PLA2G2A, PTGIS	
SMP00289	Diflunisal Pathway	1.34E–04	2	PLA2G2A, PTGIS	
SMP00077	Piroxicam Pathway	1.34E–04	2	PLA2G2A, PTGIS	
SMP00085	Ketoprofen Pathway	1.34E–04	2	PLA2G2A, PTGIS	
SMP00086	Ibuprofen Pathway	1.34E–04	2	PLA2G2A, PTGIS	
SMP00094	Sulindac Pathway	1.34E–04	2	PLA2G2A, PTGIS	
SMP00093	Diclofenac Pathway	1.34E–04	2	PLA2G2A, PTGIS	
SMP00098	Ketorolac Pathway	1.34E–04	2	PLA2G2A, PTGIS	
1457777	Antimicrobial peptides	1.45E–04	4	S100A8, PLA2G2A, DEFB1, GNLY	
SMP00102	Bromfenac Pathway	1.78E–04	2	PLA2G2A, PTGIS	
M22072	Alternative Complement Pathway	2.28E–04	2	C3, C7	
1269241	Complement cascade	3.52E–04	3	CFH, C3, C7	
M39649	Complement and Coagulation Cascades	4.57E–04	3	CFH, C3, C7	
M4732	Lectin Induced Complement Pathway	4.91E–04	2	C3, C7	
M7146	Classical Complement Pathway	6.59E–04	2	C3, C7	
M16894	Complement and coagulation cascades	7.24E–04	3	CFH, C3, C7	
1427857	Regulation of TLR by endogenous ligand	7.52E–04	2	S100A8, LY96	
M39770	Epithelial to mesenchymal transition in colorectal cancer	7.92E–04	4	FZD7, TMPRSS4, CLDN11, CLDN3	
M3952	Cells and Molecules involved in local acute inflammatory response	8.51E–04	2	C3, C7	
M39733	Cells and Molecules involved in local acute inflammatory response	8.51E–04	2	C3, C7	
SMP00083	Acetylsalicylic Acid Pathway	8.51E–04	2	PLA2G2A, PTGIS	
PW:0000485	eicosanoids metabolic	9.56E–04	2	PLA2G2A, PTGIS	
MAP00590	MAP00590 Prostaglandin and leukotriene metabolism	1.07E–03	2	PLA2G2A, PTGIS	
83073	Complement and coagulation cascades	1.07E–03	3	CFH, C3, C7	
M917	Complement Pathway	1.18E–03	2	C3, C7	
SMP00106	Meloxicam Pathway	1.18E–03	2	PLA2G2A, PTGIS	
SMP00116	Valdecoxib Pathway	1.18E–03	2	PLA2G2A, PTGIS	
SMP00084	Etodolac Pathway	1.18E–03	2	PLA2G2A, PTGIS	
SMP00087	Rofecoxib Pathway	1.18E–03	2	PLA2G2A, PTGIS	
SMP00096	Celecoxib Pathway	1.18E–03	2	PLA2G2A, PTGIS	
M39502	Complement Activation	1.43E–03	2	C3, C7	
M39398	Allograft Rejection	1.52E–03	3	C3, GNLY, C7	
M264	Endogenous TLR signaling	1.71E–03	2	S100A8, LY96	
1474301	IL-17 signaling pathway	1.72E–03	3	S100A8, CCL2, CXCL2	
M39382	Eicosanoid Synthesis	2.00E–03	2	PLA2G2A, PTGIS	
M39581	Human Complement System	2.06E–03	3	CFH, C3, C7	
M40056	16p11.2 distal deletion syndrome	2.16E–03	2	C3, PLN	
PW:0000138	Vitamin B6 metabolic	2.58E–03	1	AOX1	
1270236	Tight junction interactions	2.84E–03	2	CLDN11, CLDN3	
M39341	Spinal Cord Injury	3.47E–03	3	PLA2G2A, CCL2, CXCL2	
M11355	Tight junction	4.64E–03	3	CLDN11, CLDN3, MYH11	

PPI networks construction and validation of predictive feature biomarkers

We uploaded a list of proteins associated with DEGs, removed the disconnected nodes and then acquired a PPI network using Cytoscape software (Fig. 3A). There were 26 nodes and 50 edges on this network.

Figure 3 Screening diagnostic biomarker candidates for endometriosis.

(A) A PPI network of DEGs. (B) A plot of biomarkers selection by LASSO regression algorithm. (C) The receiver operating characteristic (ROC) curve of FMO1, PDLIM3, FAM129A, CLDN11, C3, TMPRSS4 and DEFB1 in the metadata cohort. (D) ROC curve of FMO1, PDLIM3, FAM129A, CLDN11, C3, TMPRSS4 and DEFB1 in the GSE120103 dataset.

Furthermore, the LASSO algorithm was used to narrowed down the DEGs, leading to the identification of seven variables as specific biomarkers for endometriosis (Fig. 3B). The seven genes were FMO1, PDLIM3, FAM129A, CLDN11, C3, TMPRSS4 and DEFB1. And then, we used ROC curve to estimate the predictive value of genes. As shown in Fig. 3C, the ability of PDLIM3 in discriminating endometriosis from the control samples demonstrated a favorable diagnostic value, with an AUC of 0.955 (95% CI [0.894–0.997]). Moreover, this result was confirmed in the GSE120103 dataset with an AUC of 0.836 (95% CI [0.691–0.948]). Other six genes showed no significant difference in the GSE120103 dataset (Fig. 3D).

At the same time, in order to generate more accurate and reliable results, the GSE120103 dataset was employed to verify the expression levels of the seven features. The expression levels of PDLIM3 in endometriosis tissue were notably higher than those in the normal tissue (Fig. 4A). There was no significant difference between the two groups in other six gene expression (Fig. 4A). Therefore, PDLIM3 became one of the potential specific biomarkers in endometriosis.

Figure 4 Validation of the expression of diagnostic biomarkers.

(A) The expression level of FMO1, PDLIM3, FAM129A, CLDN11, C3, TMPRSS4 and DEFB1 in the GSE120103 dataset. (B) The mRNA expression level of PDLIM3 in endometriosis and control samples. (C) The protein expression level of PDLIM3 in endometriosis and control samples. Mean ± SD, ***P < 0.001.

QRT-PCR and western blot

To validate the expression of PDLIM3, qRT-PCR and western blot were used to measure transcriptional and protein expression in both ovarian endometriosis and normal eutopic endometrium tissue from endometriosis-free women. We found PDLIM3 had higher expression level in endometriosis group than normal control (qRT-PCR: n = 6 vs. 6; Western blot: n = 7 vs. 4) (Figs. 4B and 4C). The patients of endometriosis ranged in age from 26 to 43 years (average age 32 years). The BMI was 21.76 ± 2.42 (mean ± SD). And, their rASRM endometriosis stage was III or IV. Normal group samples were from non-endometriosis patients, ranged in age from 31 to 41 years (average age 34 years). The BMI was 21.23 ± 1.21. There was no significant different in their general characteristics.

Correlation analysis between PDLIM3 and infiltrating immune cells

The composition of immune cells in endometriosis tissues and normal control tissues were shown in Fig. 5A. The proportions of naive B cells (P < 0.001), plasma cells (P = 0.012), CD8+ T cells (P = 0.013), follicular helper T cells (P < 0.001), regulatory T cells (Tregs) (P = 0.004), resting NK cells (P = 0.037), and activated NK cells (P < 0.001) in endometriosis tissues were significantly lower than those in control tissues. However, the proportions of CD4+ memory resting T cells (P < 0.001), gamma delta T cells (P = 0.031), M1 Macrophages (P = 0.020), M2 Macrophages (P < 0.001), and neutrophils (P = 0.002) in endometriosis tissues were significantly higher than those in normal tissues (Fig. 5B). However, in GSE120103, only the proportion of Tregs cell (P = 0.037) in endometriosis tissues were significantly lower than that in control tissues (Fig. S1).

Figure 5 Distribution and visualization of immune cell infiltration.

(A) The composition of immune cells in endometriosis tissues and normal control tissues. (B) Comparison of 22 immune cell subtypes between endometriosis tissues and normal tissues. Blue and red colors represent normal and endometriosis samples, respectively. (C) Correlation matrix of all 22 immune cell subtype compositions. Both horizontal and vertical axes demonstrate immune cell subtypes. Immune cell subtype compositions (higher, lower, and same correlation levels are displayed in red, blue, and white, respectively).

The association of 22 types of immune cells was calculated and drawn in Fig. 5C. Then, we found PDLIM3 was positively correlated with M2 Macrophages (r = 0.70, P < 0.001), neutrophils (r = 0.45, P < 0.001), CD4+ memory resting T cells (r = 0.44, P < 0.001), gamma delta T cells (r = 0.34, P = 0.0088), resting mast cells (r = 0.31, P = 0.019) and M1 Macrophages (r = 0.28, P = 0.037). However, PDLIM3 was negatively correlated with follicular helper T cells (r = –0.66, P < 0.001), activated NK cells (r = –0.46, P < 0.001), CD8+ T cells (r = –0.41, P = 0.0015), Tregs cells (r = –0.39, P = 0.0028), naive B cells (r = –0.37, P = 0.0047), plasma cells (r = –0.31, P = 0.018) and resting NK cells (r = –0.29, P = 0.031) (Fig. 6). Besides, PDLIM3 was significantly negatively correlated with activated mast cells (r = –0.69, P = 0.038) in GSE120103 (Fig. S1).

Figure 6 Correlation between PDLIM3 and infiltrating immune cells in endometriosis.

(A) Correlation between PDLIM3 and single type of immune cell. (B) A comprehensive correlation analysis of PDLIM3 and infiltrating immune cells in endometriosis.

Discussion

The most widly accepted theory about pathogenesis of endometriosis is retrograde menstruation and endometrial implantation reported by Sampson et al. However, it cannot perfectly explain all aspects of endometriosis. More evidence determined a combination of genetic or epigenetic alterations leads to endometriosis (Saha et al., 2015; Dalsgaard et al., 2013; Bulun et al., 2019; Manolio et al., 2009). Therefore, it is critical to acquire a clear understanding of the molecular mechanisms underlying endometriosis if we are to improve the diagnosis and treatment of endometriosis. The main purpose of our study was to detect the potential biomarker of endometriosis for the mechanism and treatment.

In this investigation, we compared three public datasets and a total of 42 DEGs were identified, including 23 upregulated genes and 19 downregulated genes. Another DEGs list of the validation set included most of 42 DEGs, which strengthen current results. The results of enrichment analyses suggested that gene function enriched by DEGs were mainly associated with immune system, such as humoral immune response, complement activation, alternative pathway, antimicrobial humoral response, cytokine–cytokine receptor interaction, systemic lupus erythematosus, chemokine signaling pathway, and regulation of complement cascade. Endometriosis is a disease with prolonged inflammatory response. The presence of ectopic tissues in peritoneal cavity, without clear mechanism, reaches a state of balance with a high level of proinflammatory cytokines and immune cells (Li et al., 2020). This evidence is consistent with our findings, proving that the results in the present study are accurate, as well as confirming that the immune response plays a vital role in endometriosis.

Based on a machine-learning algorithm, seven variables were identified from 42 DEGs and then verified their diagnosis value by ROC curve. Finally, PDLIM3 was screened out due to it is highly expressed in ectopic endometrium and has a favorable diagnostic value in all datasets. There were totally 14 peritoneal endometriosis samples, 16 ovarian endometriosis samples in three train sets and 18 ovarian endometriosis samples in the fourth verification dataset. Pelvic endometriosis includes ovarian endometriosis (OME), deep infiltrating (DIE) and superficial (SUP) endometriosis. OME is the most common lesion of endometriosis (Dalsgaard et al., 2013), which mostly accounted for our training data and totally accounted for our validating data. Therefore, we believed four datasets used in our study could effectively reflect the clinical proportions of endometriosis of different types. On the other hand, indeed, the origin of samples, which was similar but not identical, between meta-analysis and validation could partly explain the no significant different gene expression for six of the seven genes initially identified.

PDLIM3, also known as ALP (actin-associated LIM protein), is a member of the PDZ-LIM domain family (Xia et al., 1997). It is mainly contained a N-terminal PDZ domain and a C-terminal LIM domain. Previous studies revealed PDLIM3 was mainly expressed in skeletal muscle (Passier, Richardson & Olson, 2000) and could regulate skeletal muscle development (Pomiès et al., 2007; Ohsawa et al., 2011). Lak et al. (2021) identified PDLIM3 was one of the rhabdomyosarcoma markers and associated with clinical outcome. Wang et al. (2019) found that polymorphisms in PDLIM3 gene increased risk of idiopathic dilated cardiomyopathy. Other studies indicated PDLIM3 was also correlated to hypertrophic cardiomyopathy (Lopes et al., 2015; Bagnall, Yeates & Semsarian, 2010), atrial fibrillation (Vad et al., 2020), myofibrillar myopathy (Williams et al., 2021), the prognosis of hepatocellular carcinoma (Wu et al., 2021), early regional metastasis of tongue cancer (Lee et al., 2021), muscle-invasive bladder urothelial carcinoma (Feng et al., 2020) and medulloblastoma (Shou et al., 2015). It is the first time that PDLIM3 has been linked to endometriosis. We believe it will play an essential role in better understanding endometriosis mechanism and acting a potential target of therapy.

Furthermore, we used CIBERSORT to assess the types of immune cell infiltration in endometriosis and normal samples. A variety of immune cell subtypes were potentially related to the occurrence and development of endometriosis. We performed correlation analysis between PDLIM3 and immune cells, PDLIM3 was found to be correlated with M2 macrophages, neutrophils, CD4+ memory resting T cells, gamma delta T cells, M1 Macrophages, resting mast cells, follicular helper T cells, activated NK cells, CD8+ T cells, Tregs cells, naive B cells, plasma cells and resting NK cells. And, we analyzed the immune microenvironment of validation dataset. The tissue compositions and sample size of the meta-analysis and GSE120103 were different, which may be able to explain the different results. Notably, these two datasets both showed the specificity of the proportion of Tregs cells. Previous studies indicated the alteration of Tregs cell and Th17 cell may lead to the occurrence and development of ectopic endometrium implantation and accelerate the progression of endometriosis towards advanced stage (Vallvé-Juanico, Houshdaran & Giudice, 2019; Khan et al., 2019; Le et al., 2021).

Endometriosis is more than a pelvic disease. Changes in multiple systems, such as cardiovascular system, immune system and nervous system, have been found in patients with endometriosis (Taylor, Kotlyar & Flores, 2021). The shifts in circulating immune cell populations as well as the presence of proinflammatory cytokines create a widespread inflammatory environment. A precise control over various types of immune cells is demanded to achieve a safe and effective treatment on endometriosis. The novel biomarkers of endometriosis correlated with the magnitude of immune cell infiltration may be targets of therapy or predictors of therapeutic response.

The limitations of this study should also be acknowledged. First, the study was retrospective; thus, important clinical information was unavailable. Second, immune cell infiltration and the functions of PDLIM3 in endometriosis were inferred by bioinformatics analysis, and future prospective studies with larger sample sizes should be employed to validate our findings. Third, the biomarker PDLIM3 in this study is derived from endometriosis tissue, which indicates that this biomarker is not suitable to be used for early clinical diagnosis. Further study should be performed to confirm the function of PDLIM3 in endometriosis.

Conclusions

In a word, our results strongly suggest that PDLIM3 plays a previously unidentified but critical role in pathogenesis and immune environment of endometriosis. Better understanding of PDLIM3 could have a highly beneficial impact on the clinical treatment and outcomes of endometriosis patients.

Supplemental Information

Supplemental Information 1 Raw data.

note: the gene "PINR-LEI" is the target gene "PDLIM3"

Click here for additional data file.

Supplemental Information 2 The immune microenvironment of GSE120103 dataset.

(A) The composition of immune cells in endometriosis tissues and normal control tissues. (B) Comparison of 22 immune cell subtypes between endometriosis tissues and normal tissues. Blue and red colors represent normal and endometriosis samples, respectively. (C) A comprehensive correlation analysis of PDLIM3 and infiltrating immune cells in endometriosis. (D) Correlation between PDLIM3 and activated mast cell.

Click here for additional data file.

Supplemental Information 3 The statistical metrics for key differentially expressed genes (DEGs).

Click here for additional data file.

All the authors are thankful to Shanghai First Maternity and Infant Hospital and Tongji University for successful completion of this research.

Additional Information and Declarations

Competing Interests

Author Contributions

Human Ethics

Data Availability

The authors declare that they have no competing interests.

Lei Gan conceived and designed the experiments, performed the experiments, analyzed the data, prepared figures and/or tables, authored or reviewed drafts of the paper, and approved the final draft.

Jiani Sun conceived and designed the experiments, performed the experiments, analyzed the data, prepared figures and/or tables, authored or reviewed drafts of the paper, and approved the final draft.

Jing Sun conceived and designed the experiments, prepared figures and/or tables, authored or reviewed drafts of the paper, and approved the final draft.

The following information was supplied relating to ethical approvals (i.e., approving body and any reference numbers):

Ethics committee of Shanghai First Maternity and Infant Hospital.

The following information was supplied regarding data availability:

The data is available at GEO: GSE11691, GSE23339, GSE5108, GSE120103, GSE12768.

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
