# Peer review of "Bioinformatical analysis identifies PDLIM3 as a potential biomarker associated with immune infiltration in patients with endometriosis"

_PeerJ, doi:10.7717/peerj.13218_

## Round 0.1 · original submission · Major Revisions

Although the reviewers highlighted the importance of the study, they also indicated the presence of several issues. I recommend the authors to fully address them.

·

Basic reporting

This paper by Gan and al, idnetifies PDLIM3 as an interesting gene biomarker of endometriosis. The work is well performed, and convincing. The approach to use three published datasets, screen them for common genes, and building ROC curves with the samples of these articles, as well as from a fourth study is interesting and convincing.
There are I think some flaws in the English such as line 202 'All DEGs above were performed GO and reactome...' has no meaning. It should be I think something like 'The above list of DEGs was used to evaluate if they were enriched into specific GO or reactome pathways'.
The literature provided is sufficient but I found some other datasets that could be interesting to strengthen again the dataset of DEGs (for instance the dataset GSE12768 from Borghese and coworkers (2008) is almost completely consistent with the DEG list produced by the authors.
About the pictures, the small size of the characters is accessible only when enhancing the size from the images on a computer. There is some space that would allow to use a larger character size.

Experimental design

The design is I think excellent. The flow of the ideas is straightforward. The evaluation of immune cell contents presented in figure 5 is clearly an innovation in the field of endometriosis.
The fact that the gene list identified is largely confirmed by other studies is an extremely strong point. An additional point that could be given is the type of endometriosis that is under study in the datasets presented; are there ovarian (OME), deep infiltrating (DIE) or superficial (SUP) endometriosis. In addition is the fourth verification dataset from the same type of endometriosis than the others? OME was analyzed for PDLIM3 expression at the protein level (figure 4) therefore this question is important.

Validity of the findings

My feeling is that the originality and robustness of these experiment is convincing and robust. The computig of the p-values is state-of the art. The paper is concise and should be of great interest for people working in the field of endometriosis.

Additional comments

An interesting and innovative paper in the field. The statistical analysis is convincing.

·

Basic reporting

The subject of the manuscript, i.e. endometriosis, is clinically relevant and, as recently defined by Agarwal et al , “Early identification and treatment of endometriosis is essential and facilitated by a shift towards clinical diagnosis instead of relying on surgical diagnosis alone” (Am J Obstet Gynecol 2019;220: 354.e1–12). In addition, endometriosis is now defined as a chronic systemic disease and clinical challenges and novel innovations have been recently reported by Taylor et al. (Lancet 2021; 397: 839–52). On these bases, Introduction and Discussion need to be revised. In particular the major aim of the study “early clinical diagnosis” could not be based on tissue biomarkers such as those described in the present manuscript but on circulating biomarkers and comments on these aspects need to be added in the discussion.

Experimental design

On the basis of the comments in Basic Reporting, it is suggested to clarify that the comparative tissues analyses could help in identifying not only the mechanism of the disease but also potential target/s of therapy. Accordingly, it is necessary to note that pelvic endometriosis, analyzed in the meta-analysis of three datasets (GSE11691, GSE23339, GSE5108) could be different from ovarian endometriotic cysts characterized in the validation dataset (GSE120103). Indeed, the different origin of samples between meta-analysis and validation could account for the no significant different gene expression for six of the seven genes initially identified.
The case material used to validate the expression of PDLIM3 by qRT-PCR and western blot need to be better specified (number of samples and clinical characteristics of patients).

Validity of the findings

My suggestion is to highlight that:
1) The reported data could be considered a revisitation of already described data (GSE11691, GSE23339, GSE5108) that were only compared but not combined.
2) The new evaluation of the meta-analysis dataset with CIBERTSORT, tool not available when the single studies were published, enabled to confirm the strong involvement of inflammation. In this context It could be relevant to analyze also the immune microenvironment of GSE120103 dataset.

Additional comments

Please note that the abbreviation EMT is universally applied to Epithelial-Mesenchymal-Transition. It is strongly suggested to not use the abbreviation EMT for endometriosis .

---

## Round 0.2 · accepted · Accept

The authors adequately addressed reviewers' comments and now the manuscript is suitable for publication in PeerJ.

·

Basic reporting

The authors convincingly responded to my concerns.
I feel nevertheless that the text in the figure is still very small and could be further increased in size.

Experimental design

The design was good from the start. Including the additional dataset that I suggested strengthened the message.

Validity of the findings

Links between PDLIM3 and endometriosis were documented before and this paper is a strong reinforcement to this idea.

·

Basic reporting

The authors included all requested changes for both external reviewers accordingly no further comments are needed

Experimental design

The authors included all requested changes for both external reviewers accordingly no further comments are needed

Validity of the findings

YES

Additional comments

No comments